# MambaVC: Exploring Selective State Spaces for Learned Visual Compression

## Abstract

Learned visual compression is an important and active task in multimedia. Existing approaches have explored various CNN- and Transformer-based designs to model content distribution and eliminate redundancy, where balancing efficacy (*i.e.*, rate-distortion trade-off) and efficiency remains a challenge. Recently, state-space models (SSMs) have shown promise due to their long-range modeling capacity and efficiency. Inspired by this, we take the first step to explore SSMs for visual compression. We introduce MambaVC, a simple and strong compression network based on SSM. MambaVC develops a visual state space (VSS) block with a 2D selective scanning (2DSS) module as the nonlinear activation function after each downsampling, which helps to capture informative global contexts and enhances compression. On compression benchmark datasets, MambaVC achieves superior rate-distortion performance with lower computational and memory overheads. Specifically, it outperforms CNN and Transformer variants by 7.2% and 15.2% on Kodak, respectively, while reducing computation by 42% and 24%, and saving 12% and 71% of memory. MambaVC shows even greater improvements with high-resolution images, highlighting its potential and scalability in real-world applications. We also provide a comprehensive comparison of different network designs, underscoring MambaVC's advantages. Code is available at `https://anonymous.4open.science/r/MambaVC-408` and will be open-sourced.

## 1 Introduction

Visual compression is a long-standing problem in multimedia processing. In the past few decades, classical standards (Bellard, 2018; Bross et al., 2021) have dominated for a long time. With the advent of deep neural architectures like CNNs (Ballé et al., 2018; He et al., 2022; Wang et al., 2022) and Transformers (Koyuncu et al., 2022; Zou et al., 2022), learned compression methods have emerged and shown ever-improving performance, gaining increasing interest over traditional ones.

The core of visual compression is the neural network design to eliminate redundant information and capture content distribution, where it naturally presents a dilemma between rate-distortion optimization and model efficiency. While CNN-based methods (Ballé et al., 2017; Cheng et al., 2020; Duan et al., 2023; He et al., 2022; Wang et al., 2022) remain popular in many resource-limited scenarios thanks to the hardware-efficient convolution operators, their local receptive field (Luo et al., 2016) limits global context modeling capacity and thus restricts compression performance. The emergence of the Transformer as a fundamental module has brought a breakthrough to this challenge. Starting from simple early attempts(Zhu et al., 2021; Zou et al., 2022) to more advanced structural designs (Koyuncu et al., 2022; Qian et al., 2021), Transformer-based methods excel in the global perception with attention mechanisms and thereby benefit redundancy reduction. However, their quadratic complexity in computation and memory raises efficiency concerns. Although some hybrid approaches like TCM (Liu et al., 2023) combine CNNs and Transformers to balance compression efficacy and efficiency, it is not a sustainable direction for further development. Unlike previous work, we are committed to exploring promising solutions beyond engineering trade-offs toward this issue and open up fresh perspectives for future network designs.

Recently, state space models (SSMs) (Gu & Dao, 2023; Mehta et al., 2023; Wang et al., 2023), particularly the structured variants (S4) (Gu et al., 2021a), have been extensively studied. Mamba (Gu & Dao, 2023) stands out as a representative work, whose data-dependent selective mechanism

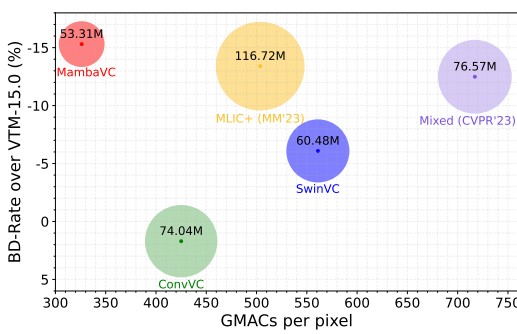 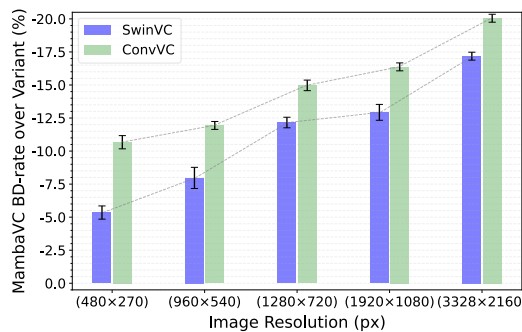

(a) BD-rate (lower is better) vs computational complexity and memory overhead (circle area) on Kodak.

(b) BD-rate of MambaVC over variants across different image resolutions on UHD (Zhang et al., 2021).

Figure 1: (a) MambaVC achieves the best BD rate with the least computation and memory overhead. See Section 4.3 and Section 4.5 for more details. (b) The improvements of MambaVC over other designs becomes more pronounced with increasing resolutions.

enhances critical information extraction while eliminating irrelevant noise from the input. This hints that Mamba-based models can effectively gather global context and thus enjoy advantages for compression. Furthermore, Mamba integrates structured reparameterization tricks and utilizes a hardware-efficient parallel scanning algorithm, assuring faster training and inference on GPUs. These compelling features inspire us to investigate Mamba's potential for visual compression.

In this paper, we introduce MambaVC, a simple and strong visual compression network with selective state spaces. Inspired by (Liu et al., 2024), we use a *visual state space* (VSS) block as the nonlinear activation function after each downsampling in the neural compression network, which integrates a specialized *2D selective scanning* (2DSS) mechanism for spatial modeling. The 2DSS performs selective scanning along 4 pre-defined traverse paths in parallel, which helps to capture comprehensive global contexts and facilitates effective and efficient compression.

We conduct extensive experiments on image and video benchmark datasets. Without the bells and whistles, MambaVC achieves a superior rate-distortion trade-off with lower computational and memory overheads compared to CNN- and Transformer-based counterparts, some as demonstrated in Figure 1(a). More encouragingly, we show that MambaVC exhibits even stronger performance on high-resolution image compression, as shown in Figure 1(b). These favorable results are consistent with SSM's efficient long-range modeling capacity, shedding light on its potential in many important yet challenging applications, such as compressing high-definition medical images and transmitting high-resolution satellite imagery. We also compare and analyze different designs from various aspects, including spatial redundancy, effective receptive field, and information loss in the compression process, to facilitate a comprehensive understanding of MambaVC's efficacy.

In summary, our contributions are as follows:

- We develop MambaVC, the first visual compression network with selective state spaces. The 2DSS improves global context modeling and helps effective and efficient compression.
- Extensive experiments on benchmark datasets show superior performance and competitive efficiency of MambaVC on image and video compression. The strong results highlight a new promising direction of compression network design beyond CNNs and Transformers.
- We showcase MambaVC's particular effectiveness and scalability in high-resolution compression, prompting its potential in many important but challenging applications.
- We compare and analyze different network designs thoroughly, showing the MambaVC's advantages regarding various aspects to validate and understand its effectiveness.

## 2 RELATED WORKS

**Learned Visual Compression** In the past decade, learned visual compression has demonstrated remarkable potential and made a significant impression. The prevailing methods can be categorized into CNN-based and Transformer-based approaches. Early works, such as CNNs with generalized divisive normalization (GDN) layers (Ballé et al., 2017; 2018; Minnen et al., 2018), achieved good performance in image compression. Attention mechanisms and residual blocks (Cheng et al., 2020;

Zhang et al., 2019; Zhou et al., 2019) were integrated into the VAE architecture later. However, the limited receptive field constrained the further development of these models. With the explosion of Vision Transformers (Dosovitskiy et al., 2020; Liu et al., 2021), Transformer-based compression models (Lu et al., 2022; Qian et al., 2021; Zhu et al., 2021; Zou et al., 2022) have shown strong competitiveness. Yet, their substantial computational and storage demands are daunting. Recent efforts (Liu et al., 2023) have attempted to combine the strengths of both approaches, but led to even increased computational complexity as shown in Figure 1(a). The trade-off between model performance and efficiency remains a pressing issue that needs to be addressed.

**State Space Models** SSMs are recently proposed models combined with deep learning to capture the dynamics and dependencies of long-sequence data. LSSL (Gu et al., 2021b) first leverages linear state space equations for modeling sequence data. Later, the structured state-space sequence model (S4) (Gu et al., 2021a) employs a linear state space for contextualization and shows strong performance on various sequence modeling tasks, especially with lengthy sequences. Building on it, numerous (Fu et al., 2022; Mehta et al., 2023; Smith et al., 2022) have been proposed, and Mamba (Gu & Dao, 2023) stands out with its data dependency and parallel scanning. Many works have consequently extended Mamba from Natural Language Processing (NLP) to the vision domain such as image classification (Liu et al., 2024; Zhu et al., 2024), multimodal Learning (Qiao et al., 2024) and others (Chen et al., 2024; Ma et al., 2024). However, the application of the Mamba for visual compression remains unexplored. In this work, we explore how to transfer the success of Mamba to build effective and efficient compression models.

## 3 METHOD

### 3.1 PRELIMINARIES: STATE-STATE MODELS AND MAMBA

State-space models (SSMs) map stimulation $x(t) \in \mathbb{R}^L$ to response $y(t) \in \mathbb{R}^L$ through a hidden state $h(t) \in \mathbb{R}^N$, where we define matrix $\boldsymbol{A}^{N \times N}$ as the evolution mapping of the hidden state, matrices $\boldsymbol{B}^{N \times 1}$ and $\boldsymbol{C}^{1 \times N}$ as the input and readout mappings for the hidden state, respectively. Typically, we can formulate the process by linear ordinary differential equations (ODEs):

$$
\begin{aligned}
h'(t) &= \boldsymbol{A}h(t) + \boldsymbol{B}x(t), \\
y(t) &= \boldsymbol{C}h(t).
\end{aligned}
\tag{1}
$$

Modern SSMs approximate this continuous-time ODE through discretization. Concretely, they discretize the continuous parameters $\boldsymbol{A}$ and $\boldsymbol{B}$ by a timescale $\Delta$, using the zero-order hold trick:

$$
\bar{\boldsymbol{A}} = \exp(\Delta\boldsymbol{A}),
\tag{2}
$$

$$
\bar{\boldsymbol{B}} = (\Delta\boldsymbol{A})^{-1}(\exp(\Delta\boldsymbol{A}) - \boldsymbol{I}) \cdot \Delta\boldsymbol{B}.
\tag{3}
$$

Then the discretized version of eq. (1) is reformulated as follows:

$$
\begin{aligned}
h_t &= \bar{\boldsymbol{A}}h_{t-1} + \bar{\boldsymbol{B}}x_t, \\
y_t &= \boldsymbol{C}h_t.
\end{aligned}
\tag{4}
$$

Mamba (Gu & Dao, 2023) further incorporates data-dependence to $\Delta$, $\boldsymbol{B}$ and $\boldsymbol{C}$, enabling an input-aware selective mechanism for better state-space modeling. While the recurrent nature restricts the fully parallel capacity, Mamba ingeniously implements structural reparameterization tricks and the hardware-efficient parallel scanning algorithm to compensate for the overall efficiency.

### 3.2 THE PROPOSED MAMBAVC

#### 3.2.1 OVERVIEW

We illustrate the architecture of MambaVC in Figure 2(a). Given an image $\boldsymbol{x}$, we first obtain the latent $\boldsymbol{y} \in \mathbb{R}^{H \times W \times C_4}$ and hyper latent $\boldsymbol{z} \in \mathbb{R}^{\frac{H}{4} \times \frac{W}{4} \times C_6}$ using the encoder $g_a$ and the hyper encoder $h_a$, respectively:

$$
\boldsymbol{y} = g_a(\boldsymbol{x}; \boldsymbol{\theta}_{g_a}),
\tag{5}
$$

$$
\boldsymbol{z} = h_a(\boldsymbol{y}; \boldsymbol{\theta}_{h_a}).
\tag{6}
$$

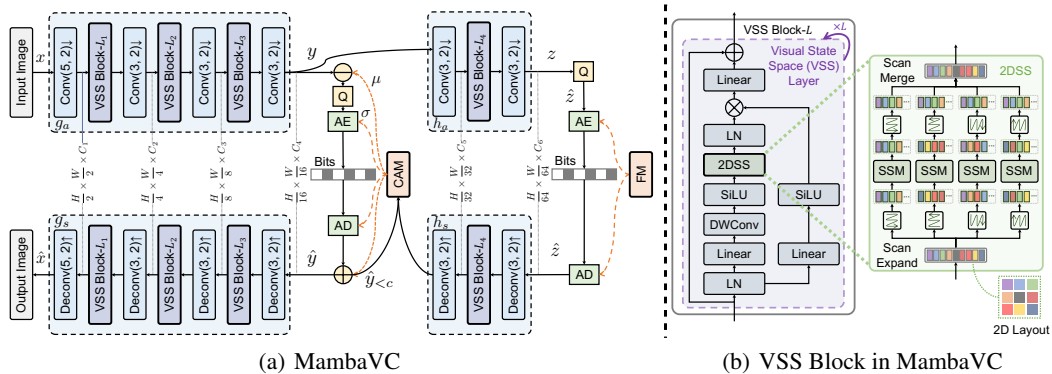

(a) MambaVC             (b) VSS Block in MambaVC

Figure 2: (a) Overview of MambaVC. **CAM** is channel-wise auto-regressive entropy model (Liu et al., 2023). **FM** is factorized entropy model. **Conv**$(N, 2) \downarrow$ and **Deconv**$(N, 2) \uparrow$ represent strided down convolution and strided up convolution with $N \times N$ filters, respectively. **AE**, **AD**, and **Q** represent Arithmetic Encoding, Arithmetic Decoding, and Quantization. (b) A VSS block consists of several layers. Each layer includes a 2DSS module, which performs selective scans in 4 parallel patterns.

Then, the quantized hyper latent $\hat{z} = Q(z)$ is entropy coded for rate $R(\hat{z}) = \mathbb{E}[-\log_2(p_{z|\psi}(\hat{z} \mid \psi))]$, where $p_{z|\psi}(\hat{z} \mid \psi) = \Pi_j \left(p_{z_j|\psi}(\psi) * \mathcal{U}\left(-\frac{1}{2}, \frac{1}{2}\right)\right)(\hat{z}_j)$, with a learned factorized prior $\psi$. $*$ denotes convolution operation.

At the decoder side, we first use a hyper decoder $h_s$ to obtain the initial mean and variance:

$$(\tilde{\boldsymbol{\mu}}, \tilde{\boldsymbol{\sigma}}) = h_s(\hat{z}; \boldsymbol{\theta}_{h_s}). \tag{7}$$

Then we divide the latent $y$ to $S$ slices $y_0, y_1, \cdots, y_{S-1}$ and compute slice-wise information by:

$$\boldsymbol{r}_i, (\boldsymbol{\mu}_i, \boldsymbol{\sigma}_i) = e_i(\tilde{\boldsymbol{\mu}}, \tilde{\boldsymbol{\sigma}}, \bar{\boldsymbol{y}}_{<i}, \boldsymbol{y}_i; \boldsymbol{\theta}_{e_i}), \tag{8}$$

$$\bar{\boldsymbol{y}}_i = \boldsymbol{r}_i + \hat{\boldsymbol{y}}_i = \boldsymbol{r}_i + Q(\boldsymbol{y}_i - \boldsymbol{\mu}_i) + \boldsymbol{\mu}_i, \tag{9}$$

where $e_i$ and $\boldsymbol{r}_i$ represent the $i$-th network and the residual in the channel-wise auto-regressive entropy model (CAM) (Liu et al., 2023), $i = 0, 1, \cdots, S-1$. We concatenate the slice-wise estimated distribution parameters and obtain the holistic $\boldsymbol{\mu}$ and $\boldsymbol{\sigma}$. We compute $R(\hat{y}) = \mathbb{E}[-\log_2(p_{\hat{y}|\hat{z}}(\hat{y} \mid \hat{z}))]$ with $p_{\hat{y}|\hat{z}}(\hat{y} \mid \hat{z}) \sim \mathcal{N}(\boldsymbol{\mu}, \boldsymbol{\sigma}^2)$.

Next, we use the decoder $g_s$ to reconstruct image from the quantized latent $\hat{y}$:

$$\hat{\boldsymbol{x}} = g_s(\hat{\boldsymbol{y}}; \boldsymbol{\theta}_{g_s}). \tag{10}$$

Finally, we optimize the following training objectives:

$$\arg\min \boldsymbol{\theta}_{g_a}, \boldsymbol{\theta}_{h_a}, \boldsymbol{\theta}_{g_s}, \boldsymbol{\theta}_{h_s}, \{\boldsymbol{\theta}_{e_i}\}_{i=0}^{S-1} \lambda \|\boldsymbol{x} - \hat{\boldsymbol{x}}\|^2 + R(\hat{z}) + R(\hat{y}), \tag{11}$$

where $\lambda$ is the Lagrangian multiplier to control the rate-distortion trade-off.

### 3.2.2 VISUAL STATE SPACE (VSS) BLOCK

Inspired by Liu et al. (2024), for the nonlinear transforms $g_a$, $g_s$, $h_a$ and $h_s$, we use a Visual State Space (VSS) block following each upsampling or downsampling operation in the middle of the transform. Figure 2(b) illustrates the structure. To be specific, each VSS Block is composed of multiple VSS layers. Following Mamba (Gu & Dao, 2023), the VSS layer adopts a gated structure with two branches. Given an input feature map $\boldsymbol{f}_{\text{in}} \in \mathbb{R}^{H \times W \times C}$, the main branch processes it by:

$$\boldsymbol{f}_{\text{hidden}} = \text{LN}_2(2\text{DSS}(\sigma(\text{DWConv}(\text{Linear}_1(\text{LN}_1(\boldsymbol{f}_{\text{in}})))))), \tag{12}$$

where LN denotes layer normalization. 2DSS denotes the 2D selective scan module, which will be elaborated in Section 3.2.3. $\sigma$ denotes the SiLU activation (Ramachandran et al., 2017). DWConv denotes the depthwise convolution. Linear denotes learnable linear projection.

Analogously, the gating branch computes the weight vector by:

$$\boldsymbol{w} = \sigma(\text{Linear}_2(\text{LN}_1(\boldsymbol{f}_{\text{in}}))). \tag{13}$$

Finally, the two branches are combined to produce the output feature map:

$$\boldsymbol{f}_{\text{out}} = \text{Linear}_3(\boldsymbol{f}_{\text{hidden}} \odot \boldsymbol{w}) + \boldsymbol{f}_{\text{in}}, \tag{14}$$

where $\odot$ denotes the element-wise product.

### 3.2.3 2D SELECTIVE SCAN (2DSS)

Vanilla Mamba (Gu & Dao, 2023) can only process 1D sequences, which can not be directly applied to 2D image data. To effectively model spatial context, we expand 4 unfolding for selective scanning. Concretely, for the feature map $\boldsymbol{f} \in \mathbb{R}^{H \times W \times C}$, where $\boldsymbol{f}[h][w] \in \mathbb{R}^C$ denotes the token in the $h$-th ($0 \le h < H$) row and $w$-th ($0 \le w < W$) column of the feature map, the unfolding patterns are defined by

$$\boldsymbol{s}_1[i] = \boldsymbol{f}[i \bmod W][\lfloor i/W \rfloor], \tag{15}$$

$$\boldsymbol{s}_2[i] = \boldsymbol{f}[(N - i - 1) \bmod W][\lfloor (N - i - 1)/W \rfloor], \tag{16}$$

$$\boldsymbol{s}_3[i] = \boldsymbol{f}[\lfloor i/H \rfloor][i \bmod H], \tag{17}$$

$$\boldsymbol{s}_4[i] = \boldsymbol{f}[\lfloor (N - i - 1)/H \rfloor][(N - i - 1) \bmod H], \tag{18}$$

where $N = H \times W$, $0 \le i < N$. $\boldsymbol{s}_1, \boldsymbol{s}_2, \boldsymbol{s}_3, \boldsymbol{s}_4 \in \mathbb{R}^{N \times C}$ are the expanded and flattened token sequences. For each flattened token sequence, we apply an S6 (Gu & Dao, 2023) operator for selective scanning, producing contextual token sequences $\boldsymbol{s}_1', \boldsymbol{s}_2', \boldsymbol{s}_3', \boldsymbol{s}_4' \in \mathbb{R}^{N \times C}$.

We then apply reversed operations to the contextual token sequences by the following folding patterns:

$$\boldsymbol{f}_1'[i][j] = \boldsymbol{s}_1'[j \times W + i], \tag{19}$$

$$\boldsymbol{f}_2'[i][j] = \boldsymbol{s}_2'[N - 1 - j \times W - i], \tag{20}$$

$$\boldsymbol{f}_3'[i][j] = \boldsymbol{s}_3'[i \times H + j], \tag{21}$$

$$\boldsymbol{f}_4'[i][j] = \boldsymbol{s}_4'[N - 1 - i \times H - j], \tag{22}$$

where $\boldsymbol{f}_1', \boldsymbol{f}_2', \boldsymbol{f}_3', \boldsymbol{f}_4' \in \mathbb{R}^{H \times W \times C}$ denote the expanded and transformed feature map of $\boldsymbol{f}$.

In the end, we merge the transformed feature maps to obtain the output feature map:

$$\boldsymbol{f}' = \boldsymbol{f}_1' + \boldsymbol{f}_2' + \boldsymbol{f}_3' + \boldsymbol{f}_4'. \tag{23}$$

### 3.2.4 EXTENSION TO VIDEO COMPRESSION

We also extend MambaVC to video compression to explore its potential. Here we choose the scale-space flow (SSF) (Agustsson et al., 2020), a renowned learned P-frame video compression model, as the base framework for extension. We upgrade the CNN-based transforms in 3 parts (*i.e.*, I-frame compression, scale-space flow, and residual) of SSF with the developed VSS blocks. We call this extension by MambaVC-SSF. We will show and discuss the experimental results in Section 4.4.

## 4 EXPERIMENTS

### 4.1 EXPERIMENTAL SETUP

#### 4.1.1 DATASETS AND TRAINING DETAILS

For image compression, we select $2 \times 10^5$ images from COCO2017(Lin et al., 2014), DIV2K(Agustsson & Timofte, 2017) and ImageNet(Russakovsky et al., 2015) as our training set. Each model is trained for 2M steps. For the first 1.2M steps, each batch consists of 8 randomly cropped 256×256 images; for the next 0.8M steps, each batch includes 2 randomly selected 512×512 upsampled images. The learning rate starts at $10^{-4}$ and drops to $10^{-5}$ at 1.8M steps, finally drops to $10^{-6}$ at 1.95M steps. We employe $\lambda \in \{0.0035, 0.0067, 0.013, 0.025, 0.05\}$ in rate-distortion loss.

For video compression, models are all trained on Vimeo-90k (Xue et al., 2019) for 1M steps at a learning rate of $10^{-4}$ and an additional 0.6M steps at $10^{-5}$. In the first phase, each batch contains 8 randomly cropped 256×256 images; in the second phase, each batch contains 8 randomly cropped 384×256 images. We optimize video model for MSE distortion metric. In particular, we use $\lambda \in \{0.00125, 0.0025, 0.005, 0.01, 0.02, 0.04, 0.08, 0.16, 0.32\}$. Inspired by (Jaegle et al., 2021; Meister et al., 2018), we process each video sequence in original and reversed order respectively during each optimization step.

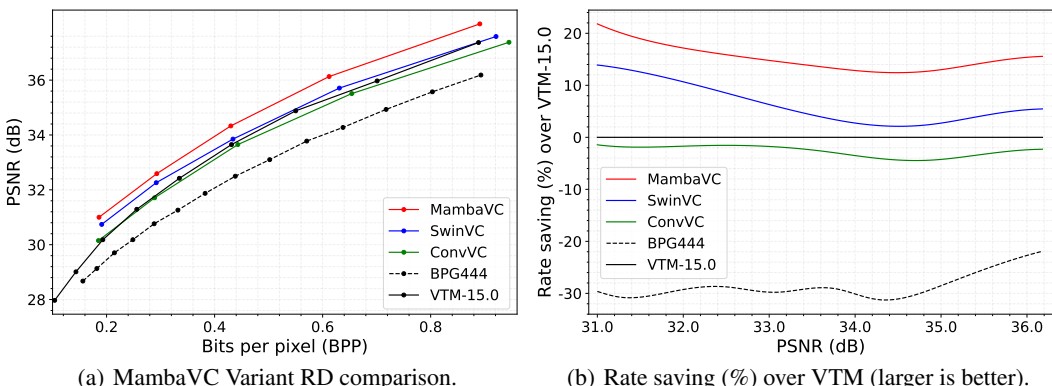

(a) MambaVC Variant RD comparison.  (b) Rate saving (%) over VTM (larger is better).

Figure 3: Comparison of compression efficiency on Kodak Franzen (1999).

### 4.1.2 BASELINES

We conduct a comprehensive and thorough evaluation of MambaVC on Kodak (Franzen, 1999), CLIC2020 (Toderici et al., 2020), JPEG-AI (JPEG-AI, 2020) and UHD (Zhang et al., 2021) with different image resolution. First, we validate the superiority of MambaVC over its convolutional and Transformer variants in terms of performance and efficiency. Specifically, we replace the VSS Block in MambaVC with swin transformer (Dosovitskiy et al., 2020) and GDN layer, respectively, naming them SwinVC and ConvVC. Detailed structures are shown in Appendix B. Secondly, we compare it with state-of-the-art methods, including both learnable and traditional methods, as presented in Appendix E.

Meanwhile, we evaluate variant SSF on MCL-JCV (Wang et al., 2016) and UVG (Mercat et al., 2020), comparing it with standard codecs AVC(x264), HEVC(x265) and the test model implementation of HEVC, called HEVC (HM). All methods fix the GOP size to 12.

### 4.2 STANDARD IMAGE COMPRESSION

The RD curves for compared image codecs on Kodak (Franzen, 1999) are shown in Figure 13(a). To provide a clearer comparison of the performance among different variants, Figure 13(b) illustrates the percentage of rate savings relative to VTM for achieving equivalent PSNR. Figure 3 demonstrates that MambaVC consistently outperforms SwinVC and ConvVC in various scenarios. SwinVC, as highlighted in previous work, surpasses ConvVC. Both MambaVC and SwinVC exhibit higher compression efficiency compared to VTM, whereas ConvVC falls short. As the rate increase, SwinVC's performance advantage slightly diminishes, while MambaVC remains unaffected.

In Table 1, we present the BD-rate of different variants compared to VTM across four datasets. MambaVC achieves an average bitrate savings of 13.35%, while SwinVC achieves an average savings of 1.94%. In contrast, ConvVC consumes an average of 4.76% more bits. Notably, MambaVC is the only variant that surpasses VTM on UHD (Zhang et al., 2021), highlighting its potential for high-resolution images, which will be discussed in the next section. Mixed (Liu et al., 2023) leverages both convolutional and Transformer structures simultaneously; however, its performance remains slightly inferior to MambaVC. See Appendix D.2 for further details.

Table 1: BD-rate (lower is better) of the variants, with VTM as the anchor.

| Method | Kodak | CLIC2020 | JPEG-AI | UHD |
|--------|-------|----------|---------|-----|
| BPG444 | 29.85% | 32.99% | 43.87% | 20.87% |
| ConvVC | 2.06% | 0.13% | 4.02% | 11.50% |
| SwinVC | -6.44% | -5.69% | -0.61% | 8.59% |
| Mixed | -12.49% | -14.36% | -10.19% | -2.16% |
| MambaVC | **-15.41%** | **-16.68%** | **-12.36%** | **-5.95%** |

The rate-distortion performance on Kodak dataset (Franzen, 1999) is shown in Figure 15. For fairness, all shown learned methods are optimized for minimizing MSE. In addition, we present the percentage

of bit savings achieved by different learning-based approaches compared to traditional methods at the same PSNR level. See Appendix E for more details.

### 4.3 HIGH-RESOLUTION IMAGE COMPRESSION

Recent work (Wang et al., 2024; Yang et al., 2024b) has demonstrated Mamba's advantages in long-range modeling. To explore this potential in visual compression, we compare our MambaVC against SwinVC and ConvVC on images of varying resolutions in two ways. Specifically, we downsample high-resolution images from the UHD (Zhang et al., 2021) by different factors to create multiple sets of images with the same distribution but different sizes. As shown in Figure 1(b), MambaVC saves more bits as the resolution increases compared to the other variants. To mitigate the impact of specific dataset distributions, we test across four datasets with different resolutions. As indicated in Table 2, the performance advantage of MambaVC on the high-resolution

Table 2: BD-rate of MambaVC over variants.

| Datasets | Mixed | SwinVC | ConvVC |
|---|---|---|---|
| Kodak | -2.01% | -7.21% | -15.25% |
| CLIC2020 | -2.24% | -13.65% | -16.02% |
| JPEG-AI | -2.05% | -11.32% | -16.08% |
| UHD | -2.53% | -17.18% | -20.05% |

Table 3: Complexity (MACs) of different models.

| Datasets | MambaVC | Mixed | SwinVC | ConvVC |
|---|---|---|---|---|
| Kodak | 0.32T | 0.71T | 0.56T | 0.42T |
| CLIC2020 | 7.24T | 15.96T | 12.45T | 9.43T |
| JPEG-AI | 7.84T | 17.48T | 13.51T | 12.52T |
| UHD | 18.02T | 35.71T | 30.98T | 23.48T |

UHD (Zhang et al., 2021) is significantly greater than on the lower-resolution Kodak (Franzen, 1999). For datasets with similar sizes, like CLIC2020 (Toderici et al., 2020) and JPEG-AI (JPEG-AI, 2020), the performance advantage is relatively consistent. MambaVC performs slightly better than Mixed and, moreover, shows a greater advantage on high-resolution datasets. We also record the change in computational cost across different resolutions. As shown in Table 3, with increasing image sizes, the computational gap widened from an initial 0.23 TMACs and 0.1 TMACs to a final 12.96 TMACs and 5.46 TMACs, separately. These results indicate that MambaVC has a distinct advantage in compressing high-resolution images. WhatMixed (Liu et al., 2023) employs a dual-branch strategy combining convolution and Transformer, introducing additional computational overhead during the separation and fusion of the two branches. As a result, its computational cost is higher than both SwinVC and ConvVC. This potential may influence the future development of specialized fields such as medical imaging and satellite imagery.

### 4.4 VIDEO COMPRESSION WITH SSF BACKBONE

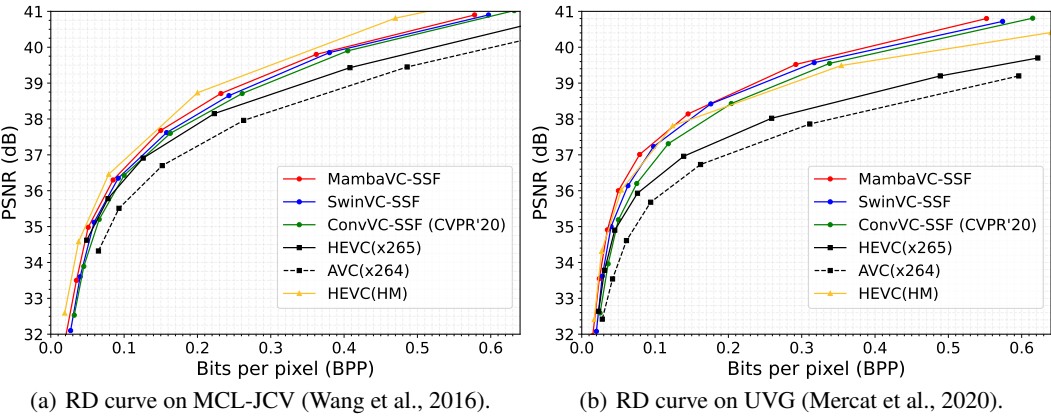

(a) RD curve on MCL-JCV (Wang et al., 2016).  (b) RD curve on UVG (Mercat et al., 2020).

Figure 4: Video compression performance evaluation on benchmark datasets.

Following the configuration of (Agustsson et al., 2020), we evaluated our method on the MCL-JCV (Wang et al., 2016) and UVG (Mercat et al., 2020) datasets. To ensure a more comprehensive comparison, we also construct the CNN- and Swin-Transformer-based counterparts with MambaVC-SSF, denoted as SwinVC-SSF and ConvVC-SSF, respectively.

Detailed configurations for different models can be found in Section 4.1.1 and appendix B. Figure 4 presents the RD curves of MambaVC-SSF with its different variants and traditional methods. Table 4 presents BD-rate with Conv-SSF model as anchor. The mamba-based model outperforms its convolutional and transformer counterparts. However, the performance improvement in video compression is not as pronounced as in image compression, possibly because merely changing the nonlinear transformation structure is insufficient to capture more redundancy.

Table 4: BD-rate of different methods compared to ConvSSF.

| Methods | MCL-JCV | UVG |
|---|---|---|
| HEVC(x265) | 25.83% | 25.97% |
| HEVC(HM) | **-24.96%** | **-15.80%** |
| SwinVC-SSF | -12.41% | -8.17% |
| MambaVC-SSF | -17.39% | -12.01% |

Additionally, all variants still fall short of HM in performance on the MCL-JCV dataset, indicating significant room for further improvement.

## 4.5 Computational and Memory Efficiencies

To explore the advantage of Mamba's linear complexity in visual compression, we evaluate the memory overhead and computational complexity on the Kodak dateset (Franzen, 1999). As results shown in Table 5, MambaVC exhibits the best performance across different variants. While MLIC+ (Jiang et al., 2023) incurs greater computational cost due to its adoption of a more advanced entropy model, it doesn't achieve superior performance. On the other hand, method (Liu et al., 2023) combining convolution and transformer, while falling short in both computational and storage aspects compared to SwinVC and ConvVC, further underscores the significance of MambaVC as a novel framework.

Table 5: Computational and memory efficiencies of different components. All models are trained with $\lambda = 0.05$. The complexity of the entropy model is attributed to the hyper decoder $h_s$. Except for (Liu et al., 2023), the other approaches have symmetric $g_a$ and $g_s$, so we do not repeat their presentation.

| Method | MACs | | | | FLOPs | | | | Peak memory | Model params |
|---|---|---|---|---|---|---|---|---|---|---|
| | $g_a$ | $h_a$ | $h_s$ | total | $g_a$ | $h_a$ | $h_s$ | total | | |
| MambaVC | 140.9G | 631.1M | 43.6G | 326.1G | 362.3G | 1.4G | 89.0G | 815.1G | 611.5M | 53.3M |
| SwinVC | 257.9G | 929.5M | 44.2G | 560.9G | 517.1G | 1.8G | 93.9G | 1.1T | 706.6M | 60.4M |
| ConvVC | 188.8G | 1.6G | 45.8G | 425.1G | 377.8G | 3.3G | 92.6G | 851.5G | 769.6M | 74.0M |
| MLIC+ (Jiang et al., 2023) | 145.9G | 1.65G | 210.2G | 503.6G | 292.1G | 3.2G | 422.5G | 1.0T | 1.3G | 116.7M |
| Mixed (Liu et al., 2023) | 267.2G | 1.0G | 46.8G | 717.1G | 544.1G | 2.2G | 90.3G | 1.5T | 877.8M | 76.6M |

## 5 Analysis

### 5.1 Latent Correlation and Distribution

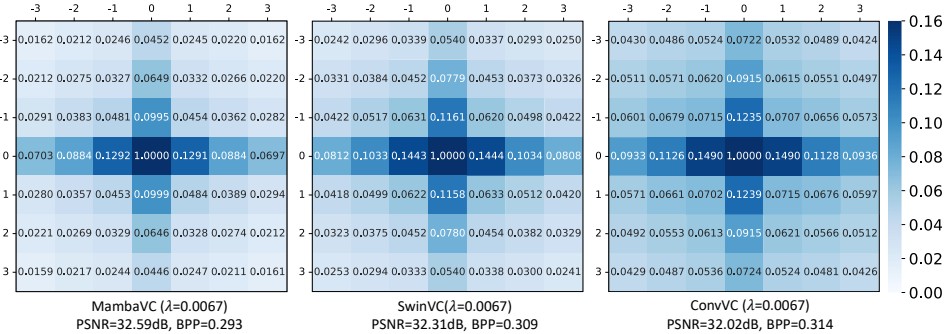

Figure 5: Latent correlation of $(\boldsymbol{y} - \boldsymbol{\mu})/\boldsymbol{\sigma}$. All models are trained with $\lambda = 0.0067$. The value at position $(i, j)$ represents cross-correlation between spatial locations $(x, y)$ and $(x + i, y + j)$ along the channel dimension, averaged across all images on Kodak (Franzen, 1999).

Learned visual compression redundancy removal involves two key steps: nonlinear encoding transform and using a conditionally factorized Gaussian prior distribution to decorrelate the latent $\boldsymbol{y}$. Specifically, the former converts the input signal from the image domain to the feature domain, while the latter uses a hyper network to learn the mean and variance $(\boldsymbol{\mu}, \boldsymbol{\sigma})$ of latent $\boldsymbol{y}$, assuming a Gaussian distribution, to further reduce correlation. As various correlations and redundancies

are eliminated, less information needs to be entropy coded, thereby improving compression efficiency. To this end, we visualized the correlation between each spatial pixel in $\ddot{\boldsymbol{y}} \triangleq (\boldsymbol{y} - \boldsymbol{\mu})/\boldsymbol{\sigma}$ and its surrounding positions, which we refer to as latent correlation. Figure 5 indicates that MambaVC has lower correlations at all distances compared to SwinVC and ConvVC. Theoretically, decorrelated latent should follow a standard normal distribution (SND). To verify this, we fit the distribution curves for different methods and calculated the KL divergence (Kullback & Leibler, 1951) from SND, as shown in Figure 6. The curve for MambaVC is noticeably closer to the SND with a smaller KL divergence (Kullback & Leibler, 1951), which indi-

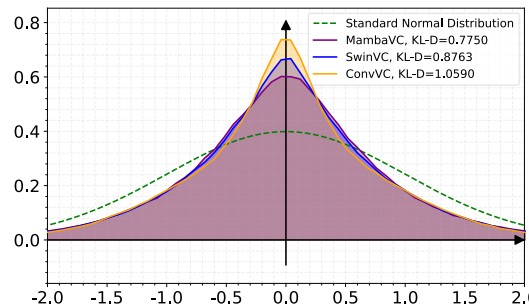

Figure 6: Distribution of $\ddot{\boldsymbol{y}}$. KL-D represents Kullback-Leibler Divergence (Kullback & Leibler, 1951) compared to the standard normal distribution.

cates the Mamba-based hyper network can learn $(\boldsymbol{\mu}, \boldsymbol{\sigma})$ more accurately. We also investigate the hyper latent correlation and the relationship between $\lambda$ and correlation, as shown in Figure 14.

## 5.2 QUANTIZE DEVIATION

In lossy compression, quantization is the primary source of information loss. We assess this loss by examining the deviation $\bar{\epsilon}$ between the latent $\boldsymbol{y} \in \mathbb{R}^{H \times W \times C}$ and its quantized counterpart $\hat{\boldsymbol{y}} \in \mathbb{R}^{H \times W \times C}$. Figure 7 presents the scaled deviation map and specific values. Each pixel in the deviation map is the mean of the absolute deviation along the channel dimension after scaling. Compared to MambaVC, SwinVC and ConvVC exhibit an average increase in information loss of 3.3% and 17%, respectively. The visualized results also indicate that MambaVC has smaller information loss at the majority of positions (deeper blue and lighter red).

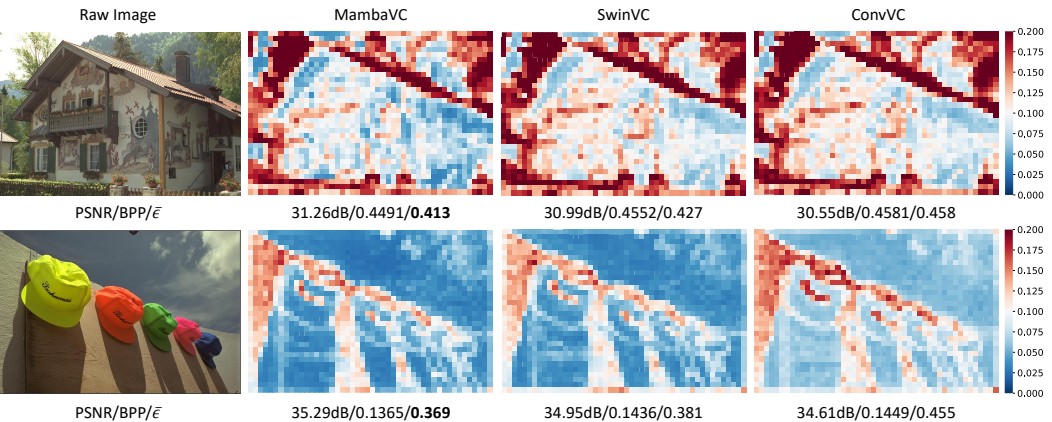

Figure 7: Scaled deviation map of *kodim03* and *kodim24* for MambaVC, SwinVC and ConvVC.

## 5.3 EFFECTIVE RECEPTIVE FIELD

The effective receptive field (ERF) (Luo et al., 2016) denotes the region of the input that a neuron in a neural network "perceives". A larger receptive field enables the network to capture related information from a wider area. This characteristic aligns perfectly with the nonlinear encoder in visual compression,

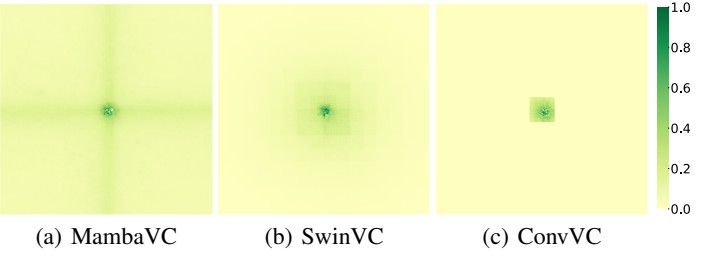

Figure 8: Effective Receptive Field (ERF) of encoders $g_a$ in different models trained on Kodak (Franzen, 1999).

as it reduces redundancy in images through feature extraction and dimensionality reduction. Consequently, we are keenly interested in examining the receptive field sizes of MambaVC and its variants. As shown in Figure 8, MambaVC is the only model with a global ERF, while ConvVC has the smallest receptive field. This confirms that in high-resolution scenarios, MambaVC can leverage more pixels globally to eliminate redundancy, whereas SwinVC and ConvVC, with their limited receptive fields, can only utilize local information, leading to performance differences.

## 6 CONCLUSIONS

In this paper, we introduced MambaVC, the first visual compression network based on the state-space model. MambaVC built a visual state space (VSS) block with 2D selective scanning (2DSS) mechanism to improve global context modeling and content compression. Experimental results showed that MambaVC achieves superior rate-distortion performance compared to CNN and Transformer variants while maintaining computational and memory efficiencies. These advantages are even more pronounced with high-resolution images, highlighting MambaVC's potential and scalability in real-world applications. Compared to other designs, MambaVC exhibits stronger redundancy elimination, larger receptive fields, and lower quantization loss, revealing its comprehensive advantages for compression. We hope MambaVC can offer a basis for exploring SSMs in compression and inspire future works.

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

## A    LIMITATIONS AND BROADER IMPACTS

### A.1    LIMITATIONS

For an objective view of our paper and to inspire future work, we discuss the limitations of MambaVC.

Instead of championing a particular implementation, this paper aims to highlight the potential of new *direction* of SSMs in visual compression, including a better performance and scalability in high-resolution Visual Compression. Without loss of generality, we use Mamba as a strong example, considering its representativeness and recent practices Liu et al. (2024); Zhu et al. (2024). Although we modify solely on SSF, we believe this approach can be extended to other CNN-based Hu et al. (2021); Li et al. (2023); Rippel et al. (2021) and Transformer-based Xiang et al. (2022) video compression models. Meanwhile, we note that there are other counterparts of Mamba, such as RWKV Peng et al. (2023) and RetNet Sun et al. (2023), or approaches like Liu et al. (2023) that effectively combine Mamba with Transformers and CNNs, which might perform better than Mamba for MambaVC. Due to the large number of SSM variants and the high computational cost of duplicate experiments, as well as the diverse methods for network fusion, we have not explored this aspect extensively.

### A.2    BROADER IMPACTS

**Positive impacts.** MambaVC enables social media platforms and video-sharing websites to upload or download data more efficiently, thereby optimizing user experience and creating a more relaxed and convenient network environment. It is also well-suited for high-resolution compression scenarios, such as medical imaging and satellite imagery, to optimize transmission efficiency.

**Negative impacts and mitigation.** Although MambaVC has reduced computational complexity and storage overhead compared to other baselines, it still imposes a computational burden on edge devices, which is a common challenge for learning-based methods. In the future, model light-weighting techniques such as network pruning, low-rank decomposition, and parameter quantization are worth exploring for application in learned compression methods.

## B    MODEL CONFIGURATIONS

### B.1    OUR METHOD

**MambaVC** The detailed architecture has been delineated in Section 3.2. For the number of channels and layers, we set them as $(C_1, C_2, C_3, C_4, C_5, C_6) = (256, 256, 256, 320, 256, 192)$ and $(L_1, L_2, L_3, L_4) = (2, 2, 9, 2)$, respectively. Due to the high resolution of images in UHD, which slows down inference, we randomly select 20 images from the UHD dataset and crop their length to 3328 pixels along the center for use as the test set.

**MambaVC-SSF** For encoder/decoder and hyper encoder/decoder in SSF Agustsson et al. (2020), there is a VSS Block following each upsampling or downsampling operation, except when generating the reconstructed image or latent with layer number $(L_1, L_2, L_3, L_4, L_5, L_6) = (1, 2, 3, 1, 1, 1)$.

### B.2    CONVOLUTIONAL VARIANT

**ConvVC** The architecture of ConvVC are shown in Figure B.1. Specifically, we replaced the VSS Block with the popular GDN layer Ballé et al. (2016), which has been proven effective in Gaussianizing the local joint statistics of natural images. To compensate for the limited effective receptive field of convolutions, we set all convolutional kernels to a size of 5. For architecture, our base model has the following parameters: $(C_1, C_2, C_3, C_4, C_5, C_6) = (448, 448, 448, 320, 448, 192)$.

### B.3    TRANSFORMER VARIANT

**SwinVC** Among a large number of vision transformer variants, we select Swin Transformer Doso-vitskiy et al. (2020) as network components for its lower complexity and superior modeling ca-

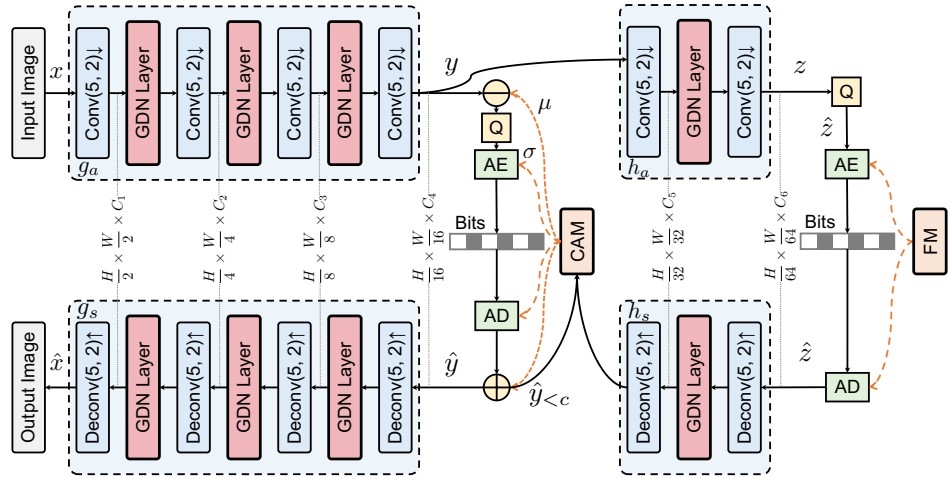

Figure 9: Architecture of ConvVC.

pability. As shown in Figure B.3, the layer number $(L_1, L_2, L_3, L_4) = (2, 2, 9, 2)$ and window size $(w_1, w_2, w_3, w_4) = (8, 8, 8, 4)$ are common to all experiments. For channels, we set $(C_1, C_2, C_3, C_4, C_5, C_6) = (256, 256, 256, 320, 256, 192)$.

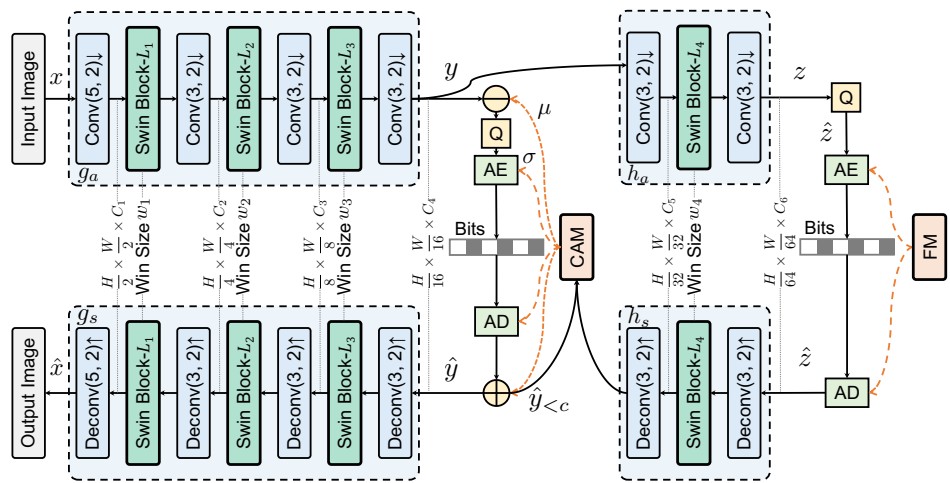

Figure 10: Architecture of SwinVC.

**SwinVC-SSF** The original downsampling modules remain untouched. Following the structure akin to the image model, we utilize the Swin Transformer (Dosovitskiy et al., 2020), albeit without any LayerNorm, instead appending a ReLU layer afterward. Both latent and hyper latent channels are set at 192. For I-frame compression, scale-space flow, and residual, we employ window sizes of 8, 4, and 8, respectively. The layer number is the same as MambaVC-SSF.

## C  CLASSICAL STANDARDS

In this section, we provide the evaluation script used for traditional methods.

### C.1  IMAGE COMPRESSION

**BPG444:** We get BPG software from `http://bellard.org/bpg/` and use command as follows:

```
bpgenc -e x265 -q [quality] -f 444
-o [encoded bitstream file] [input image file]
bpgdec -o [output image file] [encoded bitstream file]
```

**VTM:** VTM is sourced from https://vcgit.hhi.fraunhofer.de/jvet/ VVCSoftware_VTM. The command is:

```
VVCSoftware_VTM/bin/EncoderAppStatic -i [input YUV file] -c [config file]
-q [quality] -o /dev/null -b [encoded bitstream file]
-wdt 1976 -hpt 1312 -fr 1 -f 1
--InputChromaFormat=444 --InputBitDepth=8 --ConformanceWindowMode=1
VVCSoftware_VTM/bin/DecoderAppStatic -b [encoded bitstream file]
-o [output YUV file] -d 8
```

## C.2 VIDEO COMPRESSION

**AVC(x264)**

```
ffmpeg -y -pix_fmt yuv420p -s [resolution] -r [frame-rate] -crf [quality]
-i [input yuv420 raw video] -c:v libx264 -preset medium -tune zerolatency
-x264-params "keyint=12:min-keyint=12:verbose=1" [output mkv file path]
```

**HEVC(x265)**

```
ffmpeg -pix_fmt yuv420p -s [resolution] -r [frame-rate] -tune zerolatency
-y -i [input video] -c:v libx265 -preset medium -crf [quality]
-x265-params "keyint=12:min-keyint=12:verbose=1" [output file path]
```

**HEVC(HM)**

```
HM/bin/TAppEncoderStatic -c HM/cfg/encoder_lowdelay_P_main.cfg
-i [input video] --InputBitDepth=8 -wdt [width]
-hgt [height] -fr [frame-rate] -f [frames number]
-o [output video] -b [encoded bitstream file] -ip 12 -q [quality]
```

## D MORE RESULTS

### D.1 EFFECTIVE RECEPTIVE FIELD

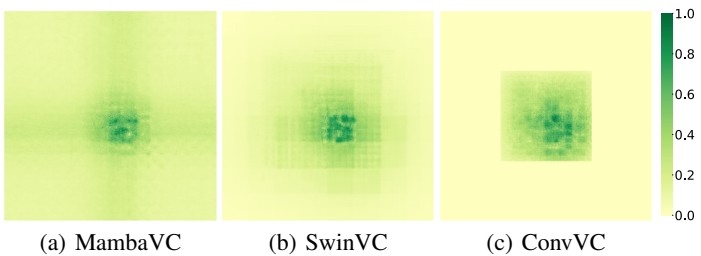

(a) MambaVC      (b) SwinVC      (c) ConvVC

Figure 11: Comparison of the ERF of the encoders and hyper encoder $g_a \circ h_a$ in MambaVC and its variants on Kodak Franzen (1999). Here, we calculate the absolute gradients $\left| \frac{d\boldsymbol{z}}{d\boldsymbol{x}} \right|$ of a pixel in the hyper latent $\boldsymbol{z}$.

In Figure 8, we present the receptive fields of latent $\boldsymbol{y}$ after passing through the encoder $g_a$. Additionally, we explore the receptive fields of the hyper latent $\boldsymbol{z}$ after passing through the hyper encoder $g_a \circ h_a$, as shown in Figure 11. Vertically comparing the methods, we observe that the receptive field expands as the network depth increases, suggesting a greater influence of surrounding areas on the value of each spatial point. Horizontally comparing the methods, MambaVC consistently demonstrates the largest receptive field among all approaches.

## D.2 VARIANT VISUAL COMPRESSION PERFORMANCE ON DIFFERENT DATASETS

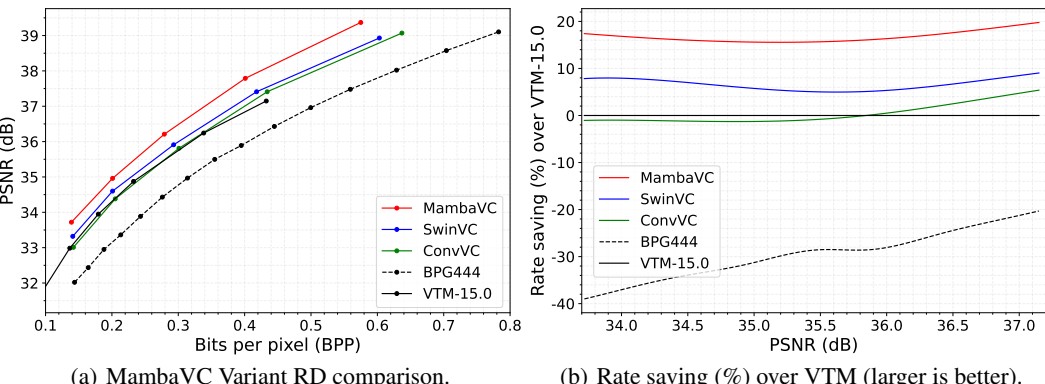

(a) MambaVC Variant RD comparison.    (b) Rate saving (%) over VTM (larger is better).

Figure 12: Comparison of compression efficiency on CLIC2020 Toderici et al. (2020) among different variants.

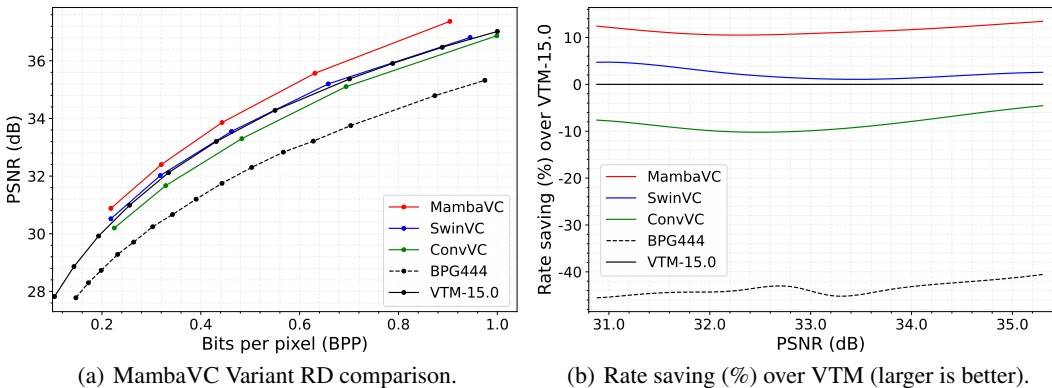

(a) MambaVC Variant RD comparison.    (b) Rate saving (%) over VTM (larger is better).

Figure 13: Comparison of compression efficiency on JPEG-AI JPEG-AI (2020) among different variants.

Additional rate-distortion results on Kodak Franzen (1999),CLIC2020 Toderici et al. (2020) and JPEG-AI JPEG-AI (2020) are shown in Figure 3, Figure 12 and Figure 13.

## D.3 INFERENCE EFFICIENCY

Table 6: Inference Efficiency for different model.

| Method | Latency(s) | | | MACs | FLOPs | Peak memory | Model params | BD-rate |
|---|---|---|---|---|---|---|---|---|
| | Encode | Decode | Total | | | | | |
| MambaVC | 0.1557 | 0.0984 | 0.2541 | 326.1G | 815.1G | 611.5M | 53.3M | -15.41% |
| SwinVC | 0.1452 | 0.1331 | 0.2783 | 560.9G | 1.1T | 706.6M | 60.4M | -6.08% |
| ConvVC | 0.1155 | 0.0911 | 0.2066 | 425.1G | 851.5G | 769.6M | 74.0M | 1.70% |
| MLIC+ | 0.1430 | 0.1224 | 0.2654 | 503.6G | 1.0T | 1.3G | 116.7M | -12.49% |
| Mixed | 0.1988 | 0.1478 | 0.3466 | 544.1G | 1.5T | 877.8M | 76.6M | -13.40% |
| FTIC | 0.1250 | 0.2420 | 0.3670 | - | - | 277.9M | 70.9M | -15.95% |

We summarize the inference storage and time overhead of each model and calculated the latency. Since the current underlying design of Mamba does not support CPU frameworks, we test the average runtime on the Kodak dataset using an RTX 4090. The results are shown in Table D.2. The actual inference speed of MambaVC is inferior to ConvVC, likely due to operations such as feature unfolding not being accounted for in the FLOPs/GMACs calculation, yet consuming substantial time during practical inference. However, the rate-distortion performance of ConvVC is much lower than MambaVC.

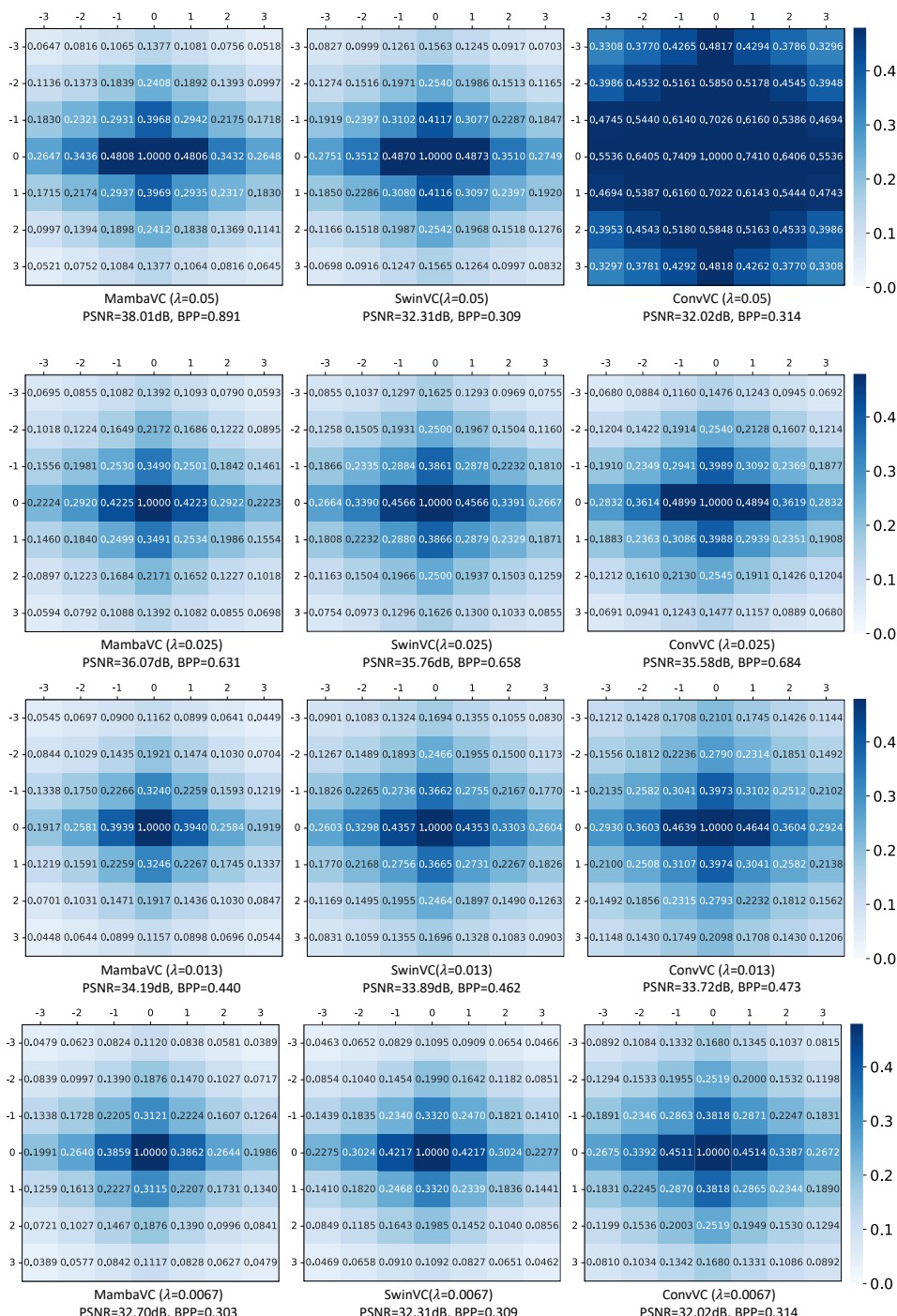

Figure 14: Latent correlation of $(\boldsymbol{z} - \mu(\boldsymbol{z}))/\sigma(\boldsymbol{z})$, averaged across all latent elements of all images on Kodak (Franzen, 1999). The value at position $(i, j)$ represents cross-correlation between spatial locations $(x, y)$ and $(x+i, y+j)$ along the channel dimension. Each row represents different variants trained with the same $\lambda$, with $\lambda$ values from top to bottom being 0.05, 0.025, 0.013, and 0.0067.

## D.4 Hyper Latent Correlation

Figure 14 illustrates the spatial correlation of the normalized prior latents. Horizontally comparing the different methods, MambaVC consistently shows the best performance across all $\lambda$. Vertically comparing the results, as the $\lambda$ decreases, the proportion of distortion loss diminishes, leading the model to focus more on compression ratio and thus eliminate more redundancy.

## D.5 THE IMPACT OF 2DSS

To validate the effectiveness of 2DSS, we select different Mamba models and various scanning strategies. First, we replace the VSS Layer, which includes the 2DSS, with the Vision Mamba Encoder Layer proposed by Zhu et al. (2024). Next, we substitute the 2D

Table 7: BD-rate compared to VTM.

| ID | Model | Kodak | CLIC2020 | JPEG-AI |
|----|-------|-------|----------|---------|
| (0) | MambaVC | -15.41% | -16.68% | -12.36% |
| (1) | Zhu et al. (2024) | -10.26% | -13.87% | -9.91% |
| (2) | Continuous 2D Scan | -14.87% | -16.02% | -12.09% |
| (3) | Bidirectional 2D Scan | -10.99% | -13.76% | -10.68% |

Scanning with the Continuous 2D Scanning method introduced in PlainMamba (Yang et al., 2024a). Finally, we modify the original four-directional scanning to a bidirectional scanning approach: horizontal and vertical, starting from the top-left to the bottom-right. Method (1) shows a significant performance gap compared to MambaVC, as the VSS Layer with 2DSS at its core outperforms the Vision Mamba Encoder Layer in Vim (Zhu et al., 2024). Compared to method (2) and (3), the number of 2D scans has a greater impact on performance than the scanning method.

# E  COMPARISON WITH THE STATE-OF-THE-ART METHODS

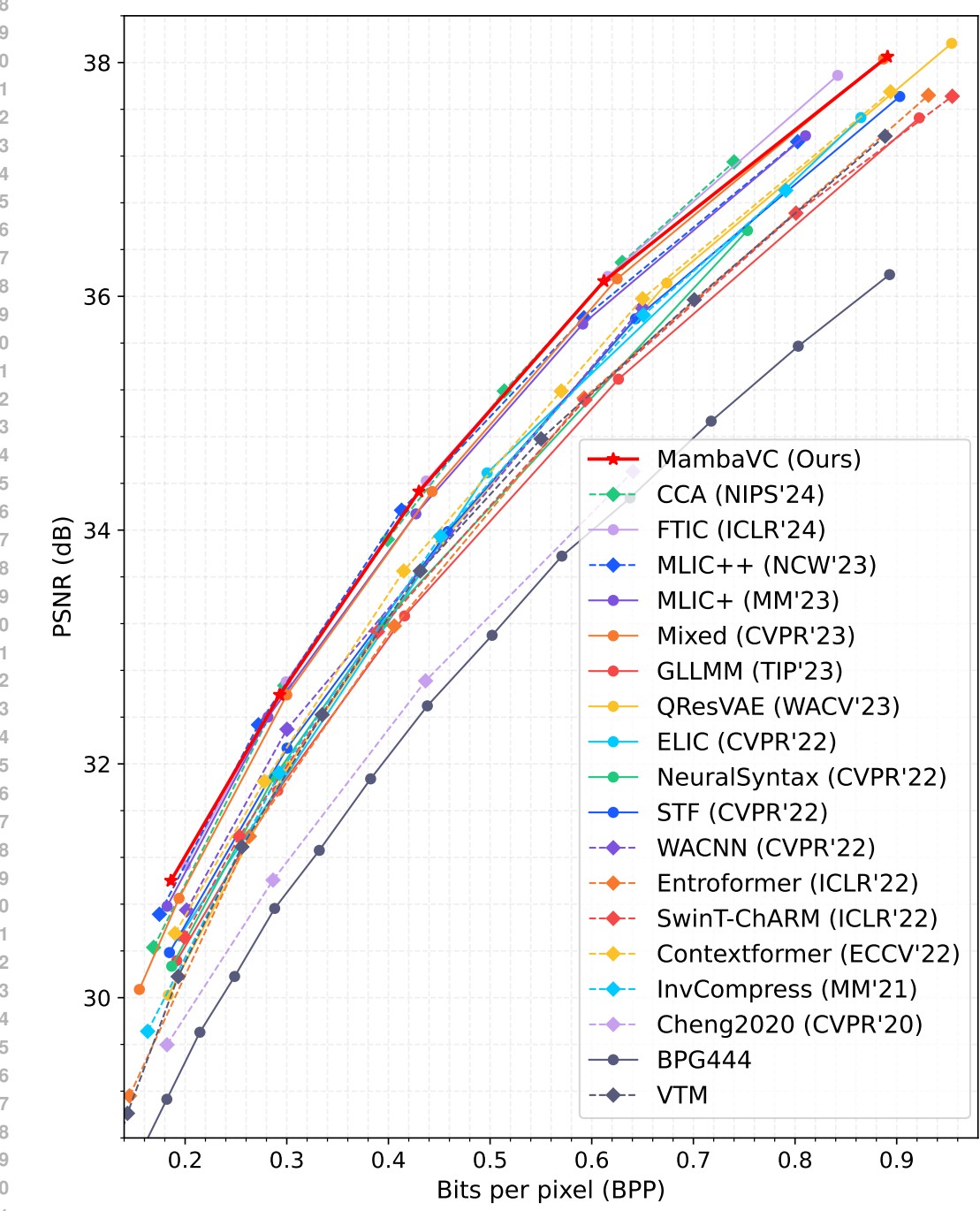

Figure 15: Rate-distortion performance on Kodak, comparing with existing works (CCA (Han et al., 2024), FTIC (Li et al., 2024), MLIC++ (Jiang et al., 2023), MLIC+ (Jiang et al., 2023), Mixed (Liu et al., 2023), GLLMM (Fu et al., 2023), QResVAE (Duan et al., 2023), ELIC (He et al., 2022), STF (Zou et al., 2022), WACNN (Zou et al., 2022), Entroformer (Qian et al., 2021), Swin-ChARM (Zhu et al., 2021), Invcompress (Xie et al., 2021), Contextformer (Koyuncu et al., 2022), NeuralSyntax (Wang et al., 2022)).

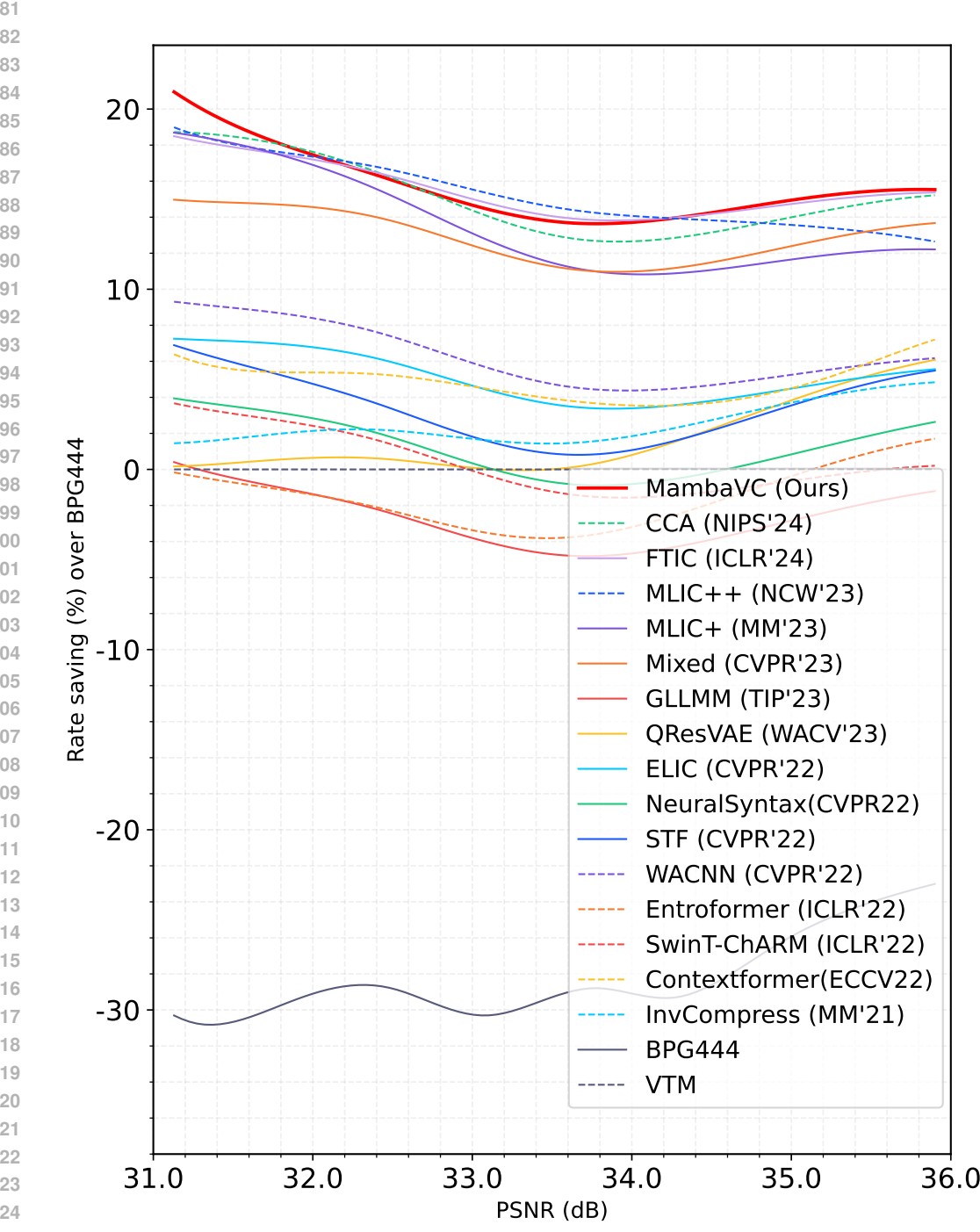

Figure 16: Percentage of rate-saving over VTM evaluated on Kodak (extended version of Figure 15), comparing with existing work (CCA (Han et al., 2024), FTIC (Li et al., 2024), MLIC++ (Jiang et al., 2023), MLIC+ (Jiang et al., 2023), Mixed (Liu et al., 2023), GLLMM (Fu et al., 2023), QResVAE (Duan et al., 2023), ELIC (He et al., 2022), STF (Zou et al., 2022), WACNN (Zou et al., 2022), Entroformer (Qian et al., 2021), Swin-ChARM (Zhu et al., 2021), Invcompress (Xie et al., 2021), Contextformer (Koyuncu et al., 2022), NeuralSyntax (Wang et al., 2022)).

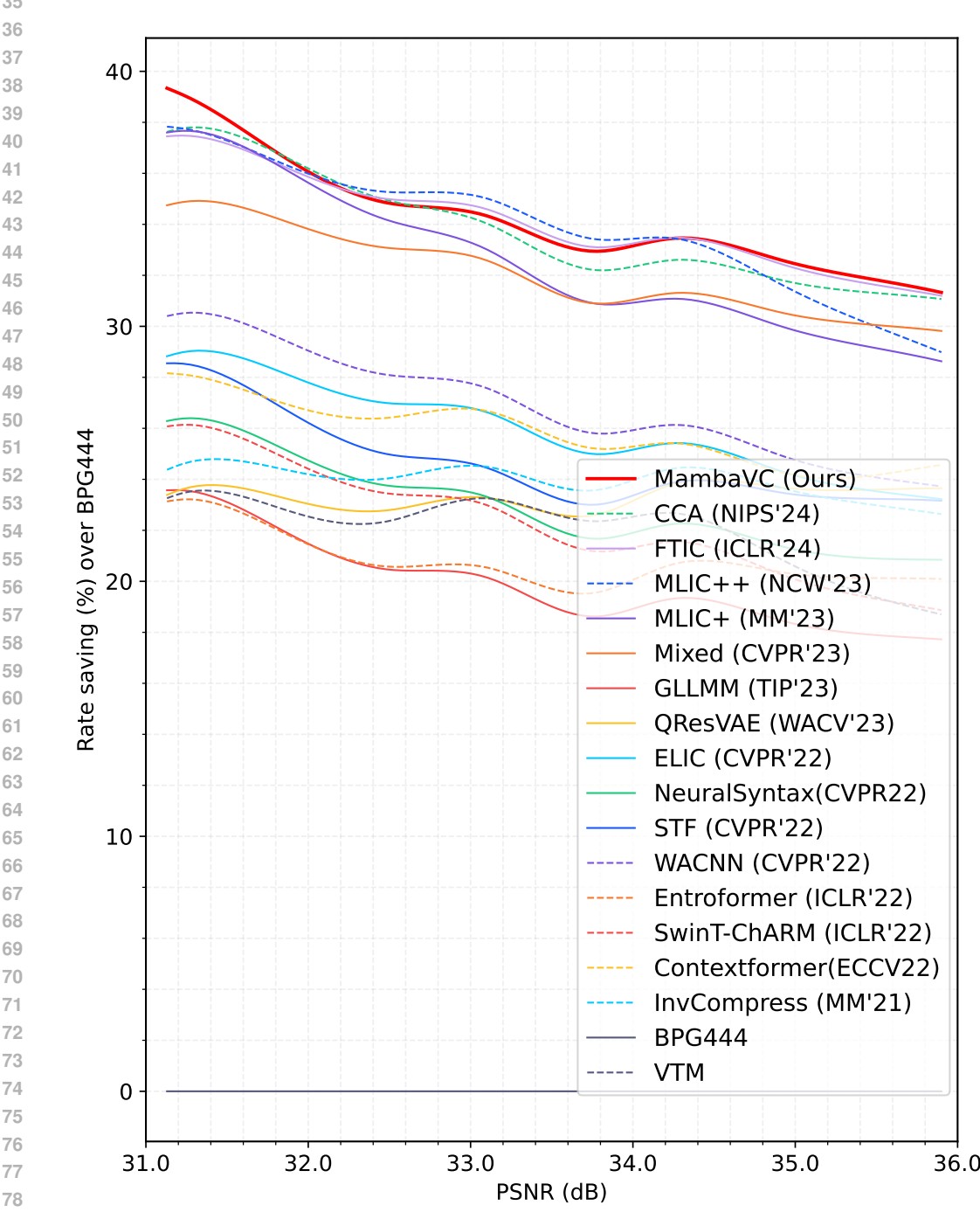

Figure 17: Percentage of rate-saving over BPG444 evaluated on Kodak (extended version of Figure 15), comparing with existing work (CCA (Han et al., 2024), FTIC (Li et al., 2024), MLIC++ (Jiang et al., 2023),MLIC+ (Jiang et al., 2023), Mixed (Liu et al., 2023), GLLMM (Fu et al., 2023), QResVAE (Duan et al., 2023), ELIC (He et al., 2022), STF (Zou et al., 2022), WACNN (Zou et al., 2022), Entroformer (Qian et al., 2021), Swin-ChARM (Zhu et al., 2021), Invcompress (Xie et al., 2021), Contextformer (Koyuncu et al., 2022), NeuralSyntax (Wang et al., 2022)).

