# OpenReview forum: "MambaVC: Exploring Selective State Spaces for Learned Visual Compression"
_ICLR.cc/2025/Conference — ICLR 2025 Conference Withdrawn Submission_

### Official Review · Reviewer_7B7G · 2024-10-30

**Soundness:** 3
**Presentation:** 3
**Contribution:** 1
**Rating:** 3
**Confidence:** 4

**Summary:**

The paper leverages state-space models (SSMs) to improve both compression performance and efficiency. Unlike traditional CNN- or Transformer-based compression models, MambaVC uses a Visual State Space (VSS) block with a 2D Selective Scanning (2DSS) module, which enhances global context modeling and reduces computational and memory requirements. Experiments demonstrate that MambaVC outperforms existing models in rate-distortion performance across multiple benchmark datasets, especially for high-resolution images.

**Strengths:**

This paper is well-organized and clearly written. Figures and tables, such as the architecture overview, BD-rate comparisons, and experimental results, are helpful in illustrating outcomes. This paper makes a contribution to the field of learned visual compression by being the first to apply State-Space Models (SSMs) for visual compression tasks.

**Weaknesses:**

1. The technical contributions of this work shown in the introduction of manuscripts are the "visual state space (VSS) block" and "2D selective scanning (2DSS) mechanism". However, these components have been already proposed in Vmamba (Liu et al., Vmamba: Visual state space model). As a result, this paper lacks technical contributions and seems more like an engineering project.

2. The comparison methods are outdated (cvpr2023), please compare the methods of the last two years, such as CCA (Causal Context Adjustment Loss for Learned Image Compression, NIPS2024).

**Questions:**

In L827, why did you choose to use “-tune zerolatency” in your video compression setup? We all know that B-frames are the key to video compression.

---

### Official Review · Reviewer_DYXP · 2024-10-30

**Soundness:** 3
**Presentation:** 2
**Contribution:** 2
**Rating:** 5
**Confidence:** 5

**Summary:**

This work proposes MambaVC, a visual compression network based on State Space Models (SSMs), balancing efficacy and efficiency in visual compression.
MambaVC introduces a Visual State Space (VSS) block to capture informative global contexts, enhancing compression performance.
MambaVC achieves superior compression efficiency compared to SwinVC and ConvVC.

**Strengths:**

This work is well-motivated.
Since current learnable image compression models are computationally costly, the paper adopts VSS blocks, a recently popular structure, to balance efficacy and efficiency in visual compression. The authors further highlight the superior performance and competitive efficiency of MambaVC in image and video compression.

**Weaknesses:**

- All experimental results of MambaVC in the main paper are only compared to customized CNN and Transformer variants (ConvVC and SwinVC), which is unreasonable.

- Fig. 15 does not provide a comparison with SoTA image compression methods, such as MLIC++[1], FTIC[2], and GroupedMixer[3]. Based on the experimental results, these methods achieve better performance than MambaVC.

- Most learned image compression methods provide experiments optimized with MS-SSIM. What performance does MambaVC achieve when optimized with this it?

- Is it possible to show all results in a table for a comparison of efficacy and efficiency metrics, such as BD-rate, latency, MACs, parameters, and FLOPs? I am not sure whether all results use the same settings, but the authors highlight that MambaVC could achieve a better balance of both efficacy and efficiency. Providing a clear comparison would improve clarity.

-  Why is changing only the nonlinear transformation structure insufficient for capturing additional redundancy? Is it influenced by the accuracy of motion estimation? Could state-of-the-art video compression methods serve as a baseline instead of SSF[4]? If even traditional compression methods like HEVC are not outperformed, does this experiment have meaningful significance? Traditional methods are likely to have a clear advantage in terms of efficiency.

- Considering the standards of ICLR, the technical contribution of this work does not appear significant, as it primarily involves the straightforward application of a VSS block on an existing visual compression baseline.

[1] MLIC++: Linear Complexity Multi-Reference Entropy Modeling for Learned Image Compression, ICMLW 2023.

[2] FTIC: Frequency-Aware Transformer for Learned Image Compression, ICLR 2024.

[3] GroupedMixer: An Entropy Model with Group-wise Token-Mixers for Learned Image Compression, TCSVT 2024.

[4] Scale-space flow for end-to-end optimized video compression, CVPR 2020

**Questions:**

Please refer to the Weaknesses for specific questions.

---

### Official Review · Reviewer_752u · 2024-10-30

**Soundness:** 2
**Presentation:** 2
**Contribution:** 2
**Rating:** 5
**Confidence:** 3

**Summary:**

The paper introduces MambaVC, an approach for learned visual compression utilizing a state space model (SSM). This method diverges from previous CNN- and Transformer-based methods.
The authors design a visual state space (VSS) block and a 2D selective scanning (2DSS) module for a better balance between compression efficiency and effectiveness. The reported experimental results demonstrate that the proposed MambaVC outperforms both ConvVC and SwinVC in terms of compression performance and efficiency, with significant reductions in computational and memory requirements.
Furthermore, the authors provide the code for further research.

**Strengths:**

1. The authors have provided anonymous code and promised the code will be released. This increases the credibility of the work and is helpful for other researchers to further investigate and build upon this work.
2. The authors demonstrate that with increasing image resolution, the absolute improvement in BD-rate for the proposed MambaVC compared to previous methods, i.e., SwinVC and ConvVC, becomes more significant.
3. The authors conducted experiments not only on image compression but also on video compression.
4. The authors display latent correlation in Figures 5 and 14, showing that the features obtained by MambaVC encode information more effectively.
5. In the appendix, the authors include a comparison with additional methods, where MambaVC achieves better performance than the compared methods.

**Weaknesses:**

1. The main comparative experiments in the paper lack persuasiveness. In the text, the primary experiments focus on comparisons with SwinVC and ConvVC. However, these methods are not current state-of-the-art (SOTA) and do not represent the best design when integrating CNNs and Transformers.
2. The authors do not compare their work with current SOTA methods. The comparisons in Figure 3 do not include newer SOTA experiments, such as work [1] published at ICLR 2024 a year ago.
3. The authors do not clearly explain the motivation for using Mamba in visual compression tasks. Compared to the changes from CNNs to Transformers, Mamba is more akin to a variant of Transformer rather than a completely different operator.
4. The proposed design for adopting Mamba in visual compression tasks lacks innovation and seems similar to the existing Vision Mamba [2].
5. In video compression experiments, the performance of the proposed method is inferior to the existing open-source official standard model, HEVC (HM).


----
1. Li, H., Li, S., Dai, W., Li, C., Zou, J., & Xiong, H. (2023). Frequency-Aware Transformer for Learned Image Compression. *arXiv preprint arXiv:2310.16387*.
2. Zhu, L., Liao, B., Zhang, Q., Wang, X., Liu, W., & Wang, X. (2024). Vision mamba: Efficient visual representation learning with bidirectional state space model. *arXiv preprint arXiv:2401.09417*.

**Questions:**

1. I am very curious about the performance and computational requirements of MambaVC relative to other state-of-the-art (SOTA) methods. I hope the authors can provide a comparative performance analysis. If a comparison is not possible, I would also appreciate an explanation of the reasons why.
2. I would like the authors to further discuss why Mamba is used in visual compression tasks. Could the authors please provide a more detailed explanation of their motivation for using Mamba specifically for visual compression? For example, what specific characteristics of visual compression align with the features of Mamba?
3. Meanwhile, I would like the authors to compare the advantages and disadvantages of using Mamba in visual compression tasks relative to CNNs and Transformers. Could the authors please discuss what properties of Mamba they believe make it well-suited for this task compared to other approaches, and how these properties align with the requirements of visual compression tasks?
4. I hope the authors can further explain the differences between the proposed module and the existing Vision Mamba work, as well as the aspects specifically designed for visual compression tasks.
5. I would like the authors to explain how MambaVC compares in terms of advantages and disadvantages to HEVC (HM) in video compression. Could the authors please provide more analysis on why MambaVC underperforms HEVC (HM) for video compression and what potential improvements could be made to address this gap in performance?


If the authors are willing to address my questions, I would be very pleased to continue this discussion with them.

**Details Of Ethics Concerns:**

Nan

---

### Official Review · Reviewer_kCv8 · 2024-11-04

**Soundness:** 3
**Presentation:** 2
**Contribution:** 3
**Rating:** 6
**Confidence:** 3

**Summary:**

This paper introduces a new visual compression network called MambaVC. Based on the State Space Model (SSM), MambaVC uses Visual State Space (VSS) blocks and a 2D Selective Scan (2DSS) module to enhance global context modeling, improving performance and efficiency in image compression.  MambaVC outperforms traditional Convolutional Neural Networks (CNN) and Transformer models in image compression benchmarks, showing clear advantages in computational and memory efficiency. The paper also presents experiments and evaluations demonstrating MambaVC's potential in high-resolution image compression, comparing it with other network designs to validate its advantages in compression efficiency, computational resource consumption, and memory usage.

**Strengths:**

1. **2DSS**: Authors introduce VSS blocks, which incorporate the 2D Selective Scan (2DSS) mechanism, enabling selective scanning along four predefined paths. This design allows the model to process spatial information in parallel, enhancing its ability to capture global context while reducing unnecessary information redundancy.

2. **Performance on Benchmarks**: Experiments on benchmark datasets show superior performance and competitive efficiency of MambaVC on image and video compression.

3. **High-Res Results**: Authors show the effectiveness and scalability to high-resolution compression

**Weaknesses:**

1. **Lack Baseline Comparision for 2DSS**: Although the authors compare MambaVC with ConvVC and SwinVC, the core innovation of the paper, 2DSS, is not analyzed in depth. It remains unclear how significant the impact of 2DSS is. The authors should compare the 2DSS mechanism with vanilla 2D inputs handling method of Mamba model, such as Vision Mamba[R1]’s approach, to demonstrate the effectiveness of 2DSS.

2. **Variations of 2DSS**: A deeper exploration of 2DSS appears to be missing. The current 2DSS uses cardinal scan patterns, but how would alternative patterns (e.g., diagonal) affect performance? Additionally, what is the impact of varying the number of scan patterns on the results?

[R1] Zhu, Lianghui, et al. "Vision mamba: Efficient visual representation learning with bidirectional state space model." arXiv preprint arXiv:2401.09417 (2024).

**Questions:**

1. **Training For Video**: Can the authors provide more details about how to adapt the image compression method to video compression? How does the quality of video datasets influence the performance?

---

### Official Review · Reviewer_zeRP · 2024-11-04

**Soundness:** 3
**Presentation:** 3
**Contribution:** 2
**Rating:** 5
**Confidence:** 5

**Summary:**

This paper is the first work that uses state-space models (SSMs) for learned image compression. The authors apply the VSS block to the encoder and decoder in the auto-regressive entropy model. Although the method is simple, it achieves strong performance compared with CNN and Transformer variants.

**Strengths:**

The method is well-motivated and reasonable. The authors conduct comprehensive experiments on standard image compression, high-resolution image compression, and video compression with SSF backbone. The paper is easy to understand.

**Weaknesses:**

The novelty of the proposed method is limited. The effectiveness of Mamba has been demonstrated on various CV tasks like image restoration (MambaIR), image recognition (VMamba), etc. The VSS block has been proposed in VMamba. Applying Mamba to learned image compression without a specific design does not meet the novelty bar of ICLR. Below are some suggestions on novelty and experiments.

[1]. For the experiments on image compression (section 4.2 and section 4.3), the authors should at least compare with LIC-TCM (Learned Image Compression with Mixed Transformer-CNN Architectures). The main difference between the proposed method and LIC-TCM is replacing the TCM block with the VSS block if I understand correctly.

[2]. To increase the novelty, the authors can consider different designs for the encoder and the decoder. In learned image compression, the encoder needs more contextual information, therefore, blocks like transformer and Mamba are more suitable for the encoder. However, the decoder needs to recover the image details, therefore, blocks like convolution might be more suitable for the decoder.

**Questions:**

see the weaknesses part.

---

### Note · Authors · 2024-11-25

I have read and agree with the venue's withdrawal policy on behalf of myself and my co-authors.